# `TAPAS`: a Toolbox for Adversarial Privacy Auditing of Synthetic Data

**Florimond Houssiau**
The Alan Turing Institute
The Office for National Statistics
fhoussiau@turing.ac.uk

**James Jordon**
The Alan Turing Institute
jjordon95@gmail.com

**Samuel N. Cohen**
University of Oxford
The Alan Turing Institute
samuel.cohen@maths.ox.ac.uk

**Owen Daniel**
The Office for National Statistics
owen.daniel@ons.gov.uk

**Andrew Elliott**
University of Glasgow
Andrew.Elliott@glasgow.ac.uk

**James Geddes**
The Alan Turing Institute
jgeddes@turing.ac.uk

**Callum Mole**
The Alan Turing Institute
The Office for National Statistics
cmole@turing.ac.uk

**Camila Rangel-Smith**
The Alan Turing Institute
crangelsmith@turing.ac.uk

**Lukasz Szpruch**
University of Edinburgh
The Alan Turing Institute
lszpruch@ed.ac.uk

## Abstract

Personal data collected at scale promises to improve decision-making and accelerate innovation. However, sharing and using such data raises serious privacy concerns. A promising solution is to produce synthetic data, artificial records to share instead of real data. Since synthetic records are not linked to real persons, this intuitively prevents classical re-identification attacks. However, this is insufficient to protect privacy. We here present `TAPAS`, a toolbox of attacks to evaluate synthetic data privacy under a wide range of scenarios. These attacks include generalizations of prior works and novel attacks. We also introduce a general framework for reasoning about privacy threats to synthetic data and showcase `TAPAS` on several examples.

## 1 Introduction

Synthetic data generation is a promising technology to enable access to sensitive data and accelerate machine learning projects while protecting user privacy, and has attracted a lot of attention from research and industry. The key idea is to produce and share semantically, and possibly statistically, accurate artificial data sets instead of the real data.

NeurIPS 2022 Workshop on Synthetic Data for Empowering ML Research.

While this intuitively seems to protect individuals, synthetic records are not automatically private. Apart from the obvious pitfall that large models can memorize real records, reproducing many statistics of the original data can enable reconstruction attacks [1, 2]. Further, in some cases, a motivated attacker can infer sensitive information about real records based only on synthetic ones [3]. Adversarial evaluation of privacy is a promising approach, which has the ability to take into account a large range of potential attackers, with different knowledge of the real dataset and synthetic data generation algorithm, and different attack goals. However, existing attacks are limited in scope, usually applying to one specific threat model.

In this paper, we present `TAPAS`, a toolbox for evaluating the privacy of synthetic data generators. We first propose a threat modeling framework to define privacy attacks. This framework models a wide range of attacker background knowledge and goals, allowing for a context-aware privacy analysis. Second, we describe the library of attacks implemented in `TAPAS`. Finally, we showcase the privacy evaluation of `TAPAS`, applying our toolbox to several attack scenarios and generators. We also highlight gaps in current research on empirical privacy evaluation. The toolbox is open source, and its code is available at `https://github.com/alan-turing-institute/privacy-sdg-toolbox`.

## 2 Background

### 2.1 Synthetic Data Generation

Denote by $\mathcal{X}$ a set of potential records, and by $\mathcal{D} = \bigcup_{N \in \mathbb{N} \cup \{0\}} \mathcal{X}^N$ the set of finite collections (with repeats) taken from $\mathcal{X}$. A *synthetic data generation model* (SDG or *generator*) is a randomized[1] function $\mathcal{G} : \Omega \times \mathcal{D} \to \mathcal{D}$ that takes as input a (*real*) dataset $D^{(r)} \in \mathcal{D}$ and produces a *synthetic* dataset $D^{(s)} = \mathcal{G}(D^{(r)})$. The goal is for the synthetic data $D^{(s)} = \mathcal{G}(D^{(r)})$ to be a useful replacement for the real data $D^{(r)}$ for given data science tasks, without revealing sensitive information about $D^{(r)}$. Generators typically operate in two steps: a *training* step, where a parameter $\theta \in \Theta$ is adapted to $D^{(r)}$, and a *sampling* step, where synthetic records are sampled i.i.d. from a distribution $p_\theta$ on $\mathcal{X}$.

### 2.2 Privacy attacks on synthetic data

Adversarial approaches can evaluate the privacy of synthetic data, either as an alternative or complement to formal guarantees. Stadler et al. [3] proposed a black-box attack against synthetic data that relies on shadow modeling. Lu et al. [4] and Yale et al. [5] evaluate privacy using the distance between synthetic and real records. In some settings, attacks against other privacy-enhancing technologies can apply to synthetic data, including Generative Adversarial Networks [6, 7]. Commonly, generators use aggregate statistics from real data when generating synthetic data. In this case, attacks developed against aggregate statistics apply, e.g. [8, 9, 10].

Adversarial approaches have three key advantages: (1) *Bug detection*: attacks can exploit bugs in implementations of theoretically secure mechanisms (e.g. leaky categories [3]); (2) *Context awareness*: attacks can incorporate assumptions on what the attacker knows, and what they try to learn; and (3) *Comparison across models*: generators can be effectively compared without requiring comparable or tight formal privacy guarantees.

### 2.3 Differential Privacy

A common approach to privacy is to ensure that the generator satisfies formal guarantees such as differential privacy (DP), introduced by Dwork et al. [11]. DP applies to randomized mechanisms $\mathcal{M} : \mathcal{D} \times \Omega \to \mathcal{S}$ and requires that distributions induced by $\mathcal{M}$ are close when datasets vary slightly. More precisely, we define two datasets to be neighbors (denoted "$\cong$") if they differ by the addition or deletion of one record.

**Definition 1** (Differential Privacy). *A randomized mechanism $\mathcal{M} : \mathcal{D} \times \Omega \to \mathcal{S}$ over datasets provides $(\varepsilon, \delta)-$Differential Privacy if, for all $d \cong d' \in \mathcal{D}$ and all $S \subset \mathcal{S}$,*

$$\mathbb{P}\left[\mathcal{M}(d) \in S\right] \leq e^\varepsilon \, \mathbb{P}\left[\mathcal{M}(d') \in S\right] + \delta.$$

---

[1]The set $\Omega$ is a probability measure space, and random variables are formally defined as functions of $\Omega$. This can be viewed as the "seed" of the randomized mechanism. For simplicity of notation, we omit the $\Omega$ argument in function calls: for $F : \Omega \times \mathcal{X} \to \mathcal{Y}$, we write $Y = F(x)$, where $Y$ is a random variable.

The parameter $\varepsilon$ is called the *privacy budget*. Smaller $\varepsilon$ implies more privacy, and typically less *utility* of the mechanism. A valuable property of differential privacy is *post-processing*: if $\mathcal{M}$ is $(\varepsilon, \delta)-$DP then, for any randomised operation $\mathcal{F} : \Omega \times \mathcal{O} \rightarrow \mathcal{O}'$, the composition $\mathcal{F} \circ \mathcal{M}$ is also $(\varepsilon, \delta)-$DP. Therefore, no operation performed on the output can "break" the privacy guarantees.

Differential privacy guarantees resistance to *membership inference attacks*, where an attacker attempts to infer whether a target record $x$ is in $D^{(r)}$. Indeed, suppose an attacker seeks to distinguish between the cases $D^{(r)} = d$ and $D^{(r)} = d' = d \cup \{x\}$, for a target $x$ and dataset $d$. Kairouz et al. [12] show a tradeoff between the attacker's true and false positive rates, which applies even in the worst-case attack model where the target $x$ and the dataset $d = D^{(r)} \setminus \{x\}$ are known.

**Theorem 1** ([12]). *Let $\mathcal{M} : \Omega \times \mathcal{D} \rightarrow \mathcal{O}$ satisfy $(\varepsilon, \delta)-$differential privacy and $d, d' \in \mathcal{D}$ be neighboring datasets ($d \cong d'$). Then, for any randomized attacker $\mathcal{A} : \Omega \times \mathcal{O} \rightarrow \{d, d'\}$,*

$$e^\varepsilon \geq \max\left(\frac{TP_\mathcal{A} - \delta}{FP_\mathcal{A}}, \frac{1 - FP_\mathcal{A} - \delta}{1 - TP_\mathcal{A}}\right),$$

*where $TP_\mathcal{A} = \mathbb{P}\left[\mathcal{A}(\mathcal{M}(d)) = d\right]$ and $FP_\mathcal{A} = \mathbb{P}\left[\mathcal{A}(\mathcal{M}(d')) = d\right]$ are true and false positive rates.*

However, this inequality is not generally tight. One can thus define the *effective* $\varepsilon^{\mathrm{eff}}(\delta)$:

$$\varepsilon^{\mathrm{eff}}(\delta; d, d') = \log \sup_{\mathcal{A}:\Omega \times \mathcal{O} \rightarrow \{0,1\}} \max\left(\frac{TP_\mathcal{A} - \delta}{FP_\mathcal{A}}, \frac{1 - FP_\mathcal{A} - \delta}{1 - TP_\mathcal{A}}\right)$$

For any $d \cong d'$, Theorem 1 shows $\varepsilon(\delta) \geq \varepsilon^{\mathrm{eff}}(\delta; d, d')$ for all $\delta \geq 0$; however $\varepsilon^{\mathrm{eff}}$ can be used to prove that a privacy analysis is tight [13, 14]. Further, $\varepsilon^{\mathrm{eff}}$ can measure privacy protection in different contexts, i.e. under different assumptions about the attacker $\mathcal{A}$. Specifically, for synthetic data, many SDGs are made differentially private with respect to their parameters $\theta$, usually through noise addition during model training. As sampling the model is a form of post-processing, $D^{(s)} = \mathcal{G}_\theta(D^{(r)})$ is equally $(\varepsilon, \delta)$-DP, but this guarantee is most likely not tight, as sampling introduces additional randomness. Thus, $\varepsilon^{\mathrm{eff}}$ can help evaluate more realistic quantitative privacy guarantees.

# 3  Adversarial toolbox

`TAPAS` is a toolbox for adversarial evaluation of synthetic data. We here introduce our threat modeling framework, which forms the foundation of the `TAPAS` analysis. We then detail the library of attacks implemented within the toolbox. We finally review the reporting options provided by `TAPAS`.

## 3.1  Threat Modeling

A threat model (or attack model) defines the setup in which attacks take place, and what an attacker is assumed to know. This defines what attacks can be performed on a synthetic dataset. The goal of formally defining a threat model is to evaluate the privacy guarantees of synthetic data *contextually*, i.e. under assumptions as to what a realistic attacker might know.

We propose a modular threat modeling framework, defined by three attributes of the attacker that can be modified independently: (1) their knowledge of the real dataset $D^{(r)}$, (2) their knowledge of the generator $\mathcal{G}$, and (3) what they are trying to learn. `TAPAS` implements this modular framework, enabling users to evaluate the privacy of generators in a wide range of scenarios within the toolbox.

### 3.1.1  Knowledge of the dataset

The attacker is assumed to have (potentially) uncertain knowledge about the real data $D^{(r)}$. This is captured by a prior over datasets, $D^{(r)} \sim \pi_D$, similarly to the framework of Li et al. [15]. Two common examples from prior work are *Auxiliary* and *Exact* knowledge. In the former, the dataset is a random subset of a larger ("population") dataset $D^{(\mathrm{pop})}$. This is a common assumption for attacker knowledge (see, e.g., [3, 16]). In the latter, the attacker knows that the dataset is one of two datasets ($d$ and $d'$) that differ only in one entry. This is typically used to verify DP guarantees [14].

`TAPAS` optimizes for targeted attacks (i.e. attacks that concern one specific record $t$) by combining prior knowledge of the dataset excluding $t$ (so $D^{(r)}_{-t} \sim \pi_{d'}$) and knowledge of the target $t$, which are

assumed to be independent from each other. For instance, if the attacker assumes $\mathbb{P}(t \in D^{(r)}) = 0.8$, and otherwise $t$ is replaced by another record $t'$, while the remainder $D_{-t}$ is sampled from $\pi'$, `TAPAS` combines these to give the prior $\pi_D : d \mapsto 0.8 \cdot I_{\{t \in d\}} \pi'(d \setminus \{t\}) + 0.2 \cdot I_{\{t' \in d\}} \pi'(d \setminus \{t'\})$.

### 3.1.2 Knowledge of the generator

It is important to model the information that the attacker has about the generator. Typically, there are three situations:

- *No-box*, where the attacker has no information about the generator.

- *Black-box*, where the attacker has exact knowledge of the generator and is able to apply the function $\mathcal{G}$ to arbitrary datasets. This is the most common assumption, as it is consistent with good security practices.

- *White-box*, which is an extension of black-box where the attacker additionally has access to trained parameters of the generators, such as model weights. This additional information can be used, e.g. to apply tailored attacks to the specific generative model used by $\mathcal{G}$.

`TAPAS` supports an additional setup, which we call *Uncertain-box*, where the generator function is parameterized by "meta-parameters" $\gamma \in \Gamma$, so $D^{(s)} = \mathcal{G}_\Gamma(D^{(r)}, \gamma)$. The attacker knows the generator function $\mathcal{G}_\Gamma$, but has uncertainty on the meta-parameter $\gamma \sim \pi_G$. `TAPAS` focuses on black-, uncertain- and no-box threat models, all three of which can be modeled as $(\mathcal{G}_\Gamma, \pi_G)$.

### 3.1.3 Attacker goal

Through their attack, the attacker aims to infer private information about the real dataset. This can be represented by a function mapping a dataset to a *decision* $g : \mathcal{D} \to \mathcal{S}$. Three common categories of attacks are:

- *Targeted Membership Inference* (MIA) for a target record $t$: assess whether the target is in the real dataset, $g : D^{(r)} \mapsto I_{\{t \in D^{(r)}\}}$.

- *Targeted Attribute Inference* (AIA) for an attribute $a$ and incomplete target record $t_{-a}$: find the value $v$ of $a$ such that the completed record $t_{-a}|v$ is in the data, $g : D^{(r)} \mapsto v$ s.t. $t_{-a}|v \in D^{(r)}$. This assumes the datasets are *tabular*, i.e. $\mathcal{X} = \mathcal{X}_1 \times \cdots \times \mathcal{X}_k$. In this paper, we focus on categorical sensitive attributes, i.e., where $|\mathcal{X}_a|$ is finite.

- *Reconstruction*: a special case where the attacker aims to reconstruct the entire real dataset $g : D^{(r)} \mapsto D^{(r)}$.

`TAPAS` currently focuses on the first two classes of attacks, as does most literature on privacy attacks on synthetic data. Reconstruction attacks are currently an underexplored research area.

## 3.2 Evaluating attacks

For a given attack goal, an attack is a (potentially randomized) function of a synthetic dataset that outputs a decision, $\mathcal{A} : \mathcal{D} \times \Omega \to \mathcal{S}$. The success of an attack is captured by a criterion $C : D^{(r)} \times \mathcal{S} \to \mathbb{R}$ for one specific dataset and decision. This can simply be whether the guess is correct $C : (D^{(r)}, s) \mapsto I_{\{g(D^{(r)})=s\}}$. The *success rate* of an attack $\mathcal{A}$, for a specific real dataset $D^{(r)^*}$ and generator meta-parameter $\gamma^*$ is $\mathbb{E}_{D^{(s)} \sim \mathcal{G}_\Gamma(D^{(r)^*}, \gamma^*)}\left[ C\left(D^{(r)^*}, \mathcal{A}(D^{(s)})\right)\right]$. In practice, `TAPAS` uses a distribution over "real" datasets $D^{(r)^*} \sim \pi_D^*$ in this evaluation, to incorporate uncertainty in $g(D^{(r)})$.

**Base Rate** The attacker's prior $\pi_D$ defines a base rate for a threat model, defined as the maximum success rate of an attack $\mathcal{A}_{\text{base}}$ that *does not have access to the synthetic dataset*: $\max_{s \in \mathcal{S}} \mathbb{E}_{D^{(r)} \sim \pi_D}[C(D^{(r)}, s)]$. In `TAPAS`, threat models are defined with a known base rate, typically $|\mathcal{S}|^{-1}$ for categorical decisions.

**Simulation**   Under a given threat model, the attacker is able to *simulate* the context, and generate *training* samples:

$$\begin{cases} D_1^{(r)}, \ldots, D_k^{(r)} \sim_{\text{i.i.d.}} \pi_D, & \gamma_1, \ldots, \gamma_k \sim_{\text{i.i.d.}} \pi_G, \\ D_i^{(s)} \sim \mathcal{G}_\Gamma\left(D_i^{(r)}, \gamma_i\right) & \forall i = 1, \ldots, k. \end{cases}$$

These samples can be used to select parameters $\theta \in \Theta$ of the attack $\mathcal{A}_\theta$, e.g. to optimize $C$ by taking $\theta^* \in \arg\max_{\theta \in \Theta} \sum_{i=1}^k C\big(D_i^{(r)}, \mathcal{A}_\theta(D_i^{(s)})\big)$.

### 3.3   Library of Attacks

`TAPAS` implements a range of attacks, or more formally, classes of attacks defined by meta-parameters. We focus on threat models with finite decision sets ($|\mathcal{S}| = n_{\text{cat}} \in \mathbb{N}$), and write without loss of generality $\mathcal{S} = \{s_1, \ldots, s_{n_{\text{cat}}}\}$. We call these "label-inference attacks". In this setup, `TAPAS` defines attacks as functions mapping a synthetic dataset to a *vector* of scores[2] $\mathcal{A} : \mathcal{D} \to \mathbb{R}^{n_{\text{cat}}}$. A decision is then based on the score, typically $\arg\max_i \sigma_i(D)$, but more complex decisions can also be considered. If $n_{\text{cat}} = 2$ ("binary attacks"), an attack can be summarized by a score and a threshold[3] as: $\mathcal{A}_{\sigma,\tau}(D) := I_{\{\sigma(D) \geq \tau\}}$.

Some attacks apply specifically to targeted membership (MIA) or attribute (AIA) inference. Targeted MIAs are binary attacks where the score is proportional to the likelihood of membership. Targeted AIAs are label-inference attacks where the decision set is the set of possible $a$ values $\{v_1, \ldots, v_l\}$.

**Shadow-Modeling Attacks**   In a shadow modeling attack, the attacker first generates a large number of "real" datasets $D_i^{(r)} \sim \pi_D$ and synthetic datasets $D_i^{(s)} = \mathcal{G}(D_i^{(r)})$, then trains a classifier $\mathcal{C}_\theta$ to predict $g(D_i^{(r)})$ from $D_i^{(s)}$. This procedure requires a classifier over *sets* $\mathcal{C}_\theta : \mathcal{D} \to \mathcal{S}$.

Stadler et al. [3] use a two-stage classifier combining a fixed set feature map $\phi : \mathcal{D} \to \mathbb{R}^l$, with a "classical" classifier $C_\theta : \mathbb{R}^l \to \mathcal{S}$ as $\mathcal{C}_\theta = C_\theta \circ \phi$. `TAPAS` implements the three feature sets of Stadler et al. (basic statistics, histograms, and correlations), as well as additional feature maps. In particular, we propose and implement the targeted counting queries feature map, $Q_{t,S} : D \mapsto \frac{1}{|D|} \sum_{x \in D} I_{\{x_s = t_s \; \forall s \in S\}}$, defined for a random subset of attributes $S$. We show in Section 4 that using this feature map empirically outperforms the feature maps of Stadler et al.

**Local Neighborhood Attacks**   Given a distance over records $\mathfrak{d} : \mathcal{X} \times \mathcal{X} \to \mathbb{R}^+$, this class of attacks uses synthetic data near a target record $t$ to make a decision. The intuition is that the neighborhood of $t$ is most likely to be influenced by the presence of $t$ in $D^{(r)}$.

Distance to closest synthetic record is a common heuristic privacy metric for synthetic data [4, 5]. It corresponds to a targeted membership inference attack with target $t$ and score $\sigma(D) = -\min_{x \in D} \mathfrak{d}(x, t)$. This generalizes the direct lookup attack ("is the real record in the data?"). Note that while this looks like a classical de-identification attack, *presence of $t$ in $D^{(s)}$ does not automatically imply a privacy violation*. This attack is an issue if and only if the record $t$ is more likely to be in $D^{(s)}$ when it is also in $D^{(r)}$.

We extend this attack to targeted attribute inference, where the score for value $v_i$ for the sensitive attribute is $\sigma_i(D) = -\min_{x \in D} \mathfrak{d}(x, t_{-a}|v_i)$.

Note that the distance $\mathfrak{d}$ can also be trained. For instance, Zhang et al. [17] propose a no-box attack against synthetic health records, using representation learning to train a similarity metric.

**Inference-on-Synthetic attacks**   This class of attacks relies on the idea that generators can "overfit" to $D^{(r)}$. In that case, it is possible to train a model $f_\theta$ to perform an attack directly on synthetic data.

For membership inference attacks, this involves fitting a density $\mathcal{L}_\theta : \mathcal{X} \to \mathbb{R}^+$ to $D^{(s)}$. This can be any statistical model, e.g. a Gaussian mixture. Some generators (e.g. [18, 19]) explicitly fit such a model, which in white-box scenarios can be used directly. Here, $\sigma(D) = \mathcal{L}_\theta(t)$.

---

[2]The $i^{\text{th}}$ entry of this vector is the score for $s_i$, a higher score corresponding to a more likely output decision.

[3]For many attacks, the score is defined independently of the data; only the threshold (or decision function) needs to be trained. Given a reasonable threshold value, these can be applied to a no-box scenario.

For attribute inference, the attacker trains a classifier $\mathcal{C}_\theta : \mathcal{X} \to \mathbb{R}^{n_{\mathrm{cat}}}$ on $D^{(s)}$ to predict the sensitive attribute $a$ from other attributes $(x_{-a})$. The score is the prediction score for the target $t_{-a}$, so $\sigma(D) = \mathcal{C}_\theta(t_{-a})$. This corresponds to a privacy metric called Correct Attribution Probability (CAP) [20]. This attack suffers from the base rate problem: correlations between sensitive and non-sensitive attributes can make this attack successful even if $t \notin D^{(r)}$. Whether this is a privacy violation is a contentious point in the community. TAPAS addresses this by randomizing the sensitive attribute of the target records uniformly at random when training the classifier.

### 3.4 Summarizing results

TAPAS provides several analyses of the outcome of attacks (called *reports*). These reports aggregate the result of a range of attacks run on many test pairs $(D_i^{(r)}, D_i^{(s)})$. Three main reports are:

- *Classification metrics*: accuracy, true/false positive rates, AUROC, as well as privacy gain metrics from Stadler et al. [3].

- *ROC curves*: For binary attacks with scores, the true positive rate vs. false positive rates for a range of thresholds.

- *Effective epsilon $\varepsilon^{\mathrm{eff}}$*: This report first greedily selects an attack and threshold on a 10% subset of testing samples, then performs the procedure of Jagielski et al. [14] to estimate a statistically significant lower bound on $\varepsilon^{\mathrm{eff}}$ using the 90% remaining samples. This can be used to empirically verify differential privacy.

## 4 Examples

In this section, we showcase how TAPAS can be used to evaluate the privacy of several generators. Our goal is to demonstrate how the toolbox can and should be used, rather than a thorough analysis of the privacy of synthetic data. Still, we show existing challenges in privacy evaluation.

We demonstrate the toolbox in a setting where privacy is particularly important, working with personal data. We apply the toolbox to the Office for National Statistics' 2011 Census Microdata Teaching File[4], which contains an anonymised random sample of 1% of responses from the 2011 Census of England and Wales, a total of ~570,000 records with 15 categorical attributes. The data is already recognised as non-disclosive, but our toolbox could be used in the future production of similar micro-data files. We consider three generators: PrivBayes [18], MST [19], and CT-GAN [21]. The first two respectively provide $\varepsilon-$ and $(\varepsilon, \delta)-$DP, while the last one does not.

*Experiment 1*: we consider an attacker performing an attribute-inference attack with auxiliary knowledge of the dataset and black-box knowledge of the generator. We split the dataset in two equal random parts (auxiliary and testing), from which "real" datasets of 5000 records are sampled. As target, we select an "outlier" record, by choosing the record with lowest log-likelihood (assuming attributes were independently drawn from their respective empirical distributions) from a random sample of 1000 records. We attack each generator independently, using $\varepsilon = 10$ and $\delta = 10^{-5}$ for DP generators. We apply a range of attacks: a local neighborhood attack with Hamming distance, an inference-on-synthetic attack with a random forest classifier, the Groundhog attack, the Groundhog attack with logistic regression as a learner, and a shadow-modeling attack with the random-query feature we propose. We compare the resulting accuracy, privacy gain and AUC of these attacks in figure 1(a).

*Experiment 2*: we demonstrate empirical estimation of $\varepsilon^{\mathrm{eff}}$ on MST [19] with $\varepsilon = 10$. We use the exact knowledge setup, where the attacker knows that the dataset is either $D^+ = d_{-t} \cup \{t\}$ or $D^- = d_{-t} \cup \{t'\}$ for an arbitrary $d$ of size 499 sampled from the full dataset, the same target $t$ and another arbitrary record $t'$. We choose this setup to detect violations (since empirical estimates of DP are computationally intensive [22]). We generate 1000 synthetic "training" datasets $D^+$ and $D^-$ to train the attacks, and 2500 synthetic "testing" datasets to evaluate $\varepsilon^{\mathrm{eff}}$. In figure 1(b), we show the ROC curves produced by TAPAS. We find that the shadow-modeling attack with random-query features outperforms all other attacks (with an AUC of 0.70, whereas the Groundhog attack achieves

---

[4]Source: Office for National Statistics licensed under the Open Government Licence v.1.0, `https://www.ons.gov.uk/census/2011census/2011censusdata/censusmicrodata/microdatateachingfile`.

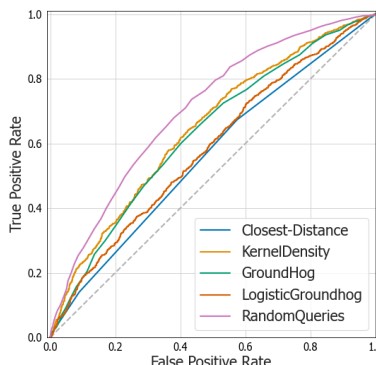

(a) Example of `TAPAS` summary statistics, in Experiment 1. The metrics are accuracy of prediction, privacy gain from [3] and AUC.

(b) ROC curve from a range of membership inference attacks in Experiment 2. The $45°$ line corresponds to a random baseline.

Figure 1: Example of outputs from `TAPAS` in the two experimental setups of Section 4

an AUC of 0.63), and is used for $\varepsilon^{\text{eff}}$ estimation. Our $\varepsilon^{\text{eff}}$ estimation procedure produces a 95% confidence interval of $\varepsilon^{\text{eff}} \in [0.86, 1.39]$. Surprisingly, $\varepsilon = 10$ is not in this confidence interval. Three main factors may be responsible for this gap: (1) the privacy analysis of the method is not tight; (2) the privacy analysis is not tight to differentiate the specific datasets $D^-$ and $D^+$ (which were sampled randomly, rather than specifically designed worst-case datasets); and (3) the attack used for $\varepsilon^{\text{eff}}$ estimation is not optimal. Further work is needed to assess which of these factors is the cause.

### Author contributions

Conceptualization: All authors; Methodology: FH, JJ, AE, JG, CM, CRS; Software: FH, JJ, JG, CM, CRS; Writing (initial): FH, SC, AE, LS; Writing (revisions): FH, OD; Project Administration: FH, JJ.

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
