# OpenReview forum: "TAPAS: a Toolbox for Adversarial Privacy Auditing of Synthetic Data"
_NeurIPS.cc/2022/Workshop/SyntheticData4ML — Neurips 2022 SyntheticData4ML_

### Official Review · Reviewer_877d · 2022-10-07
**Review of "PrivE: Empirical Privacy Evaluation of Synthetic Data Generators"**

**Rating:** 7
**Confidence:** 4

**Review:**

The authors propose a framework for empirically evaluating the privacy protection that synthetic data generation offers. Their framework implements a variety of attacks targeting different threat models and attacker goals. Given the gap between theoretical privacy guarantees and the strength of current privacy attacks, which can be massive in some cases, this kind of framework can be very useful for evaluating the privacy risk in releasing synthetic data.

The originality of the work is the collection of different attacks into a unified framework. With the exception of some novel tweaks, the attacks, threat models and metrics are from prior work.

### Pros
- The paper is clearly written.
- Collecting several attacks under one framework makes it easy for the publishers of synthetic data to evaluate privacy risks across many attacks.

### Cons
- To evaluate effective epsilon, the proposed framework only looks at a pairs of neighbouring datasets that have been sampled from a real dataset. Differential privacy is a worst-case property over all pairs of neighbouring datasets. As the worst-case dataset pairs are typically very artificial, it is likely that the effective epsilon computed with a sampled pair is a loose lower bound.
- The framework appears limited to categorical attribute inference attacks.

---

### Official Review · Reviewer_kPRj · 2022-10-11
**Interesting work**

**Rating:** 7
**Confidence:** 4

**Review:**

This paper provides a toolbox of attacks that can be used for evaluating the privacy-preserving properties of synthetic datasets. The attacks included in the toolbox are based on some previous works but have some modifications. The benefits of the work: having a framework which allows to consistently evaluate the risks of synthetic datasets is very beneficial and this paper is a nice step in that direction. It is also nice that the toolbox includes the estimation of \epsilon. Room for improvements: as you mention yourself it'd be good to test things more extensively; also for such a toolbox having the code available would be nice (but perhaps you make that available after acceptance).

---

### Official Review · Reviewer_Y1sz · 2022-10-18
**Cohesive description of practical implementations of how to approach privacy evaluation of synthetic data. Well described and strongly supported by evidence.**

**Rating:** 9
**Confidence:** 5

**Review:**

The paper presents a practical framework outlining types of attacks, threat models, and privacy evaluation metrics. The authors highlight the importance of effective epsilon (prior work) and RandomQueries (a shadow modeling approach - their novel contribution). The paper shows the benefit of using the shadow modeling approach through practical metrics.
This paper would be a great contribution to the discussion at the conference as the paper provides a concise summary of practical privacy evaluation (something the community definitely needs more of) and an easy to implement framework that compares the different evaluation methods.

The only change to propose is it would be great to see the actual numbers and the improvement through randomqueries - rather than just comparing the points on the plot. And to see, if the difference in the metrics is significantly different.

---

### Meta-Review · Area_Chair_3QSh · 2022-10-18

**Recommendation:** Accept